# Relationship between the Number of Noncommunicable Diseases and Health-Related Quality of Life in Chinese Older Adults: A Cross-Sectional Survey

**DOI:** 10.3390/ijerph17145150

**Published:** 2020-07-17

**Authors:** Jianjian Liu, Wei Yu, Jiayi Zhou, Yifan Yang, Shuoni Chen, Shaotang Wu

**Affiliations:** 1School of Health Sciences, Wuhan University, Wuhan 430071, China; jianjianliu@whu.edu.cn (J.L.); 2016302170027@whu.edu.cn (W.Y.); 2017302180144@whu.edu.cn (J.Z.); 2017302170040@whu.edu.cn (Y.Y.); 2017302170019@whu.edu.cn (S.C.); 2Global Health Institute, Wuhan University, Wuhan 430072, China

**Keywords:** noncommunicable diseases, health-related quality of life, Chinese older adults

## Abstract

China has the largest population of older adults, most of whom suffer from one or more noncommunicable diseases (NCDs). The harm of the number of NCDs on the health-related quality of life (HRQOL) of older adults should be taken seriously. A sample of 5166 adults, aged 60 years and older, was included in this study. The Chinese version of the World Health Organization Quality of Life-Old (WHOQOL-OLD) instrument was used to assess the HRQOL. Multiple linear regression models were established to determine the relationship between the number of NCDs and the total score and scores of each dimension of the WHOQOL-OLD scale. After adjusting for confounding factors, suffering from one NCD (B = −0.87, 95% CI = −1.67 to −0.08, *p* < 0.05), two NCDs (B = −2.89, 95% CI = −3.87 to −1.90, *p* < 0.001), and three or more NCDs (B = −4.20, 95% CI = −5.36 to −3.05, *p* < 0.001), all had negative impacts on the HRQOL of older adults. NCDs had significant negative impacts on the HRQOL of older adults, and as the number of NCDs increased, the HRQOL of older adults deteriorated. Therefore, we should pay attention to the prevention and management of NCDs of older adults to prevent the occurrence of multiple NCDs.

## 1. Introduction

According to the World Health Organization (WHO), the number of people over the age of 60 will reach 2 billion by 2050, which is about a fifth of the population worldwide (22%) [1]. As a country with the largest population, China has the largest number of older adults. By the end of 2018, there were 249 million (17.9%) and 167 million (11.9%) people over the age of 60 and 65, respectively [2]. However, the aging process in China is so rapid that the social system cannot keep up [1,3]. In order to address the aging crisis and ensure older adults’ right to live a long and healthy life, “healthy aging” was put forward by WHO and the Healthy China 2030 Strategy, which aimed to create an age-friendly environment through the cooperation of the whole society [4,5]. Older adults are in a critical stage of maintaining their health-related quality of life (HRQOL), where the leading health threat is noncommunicable diseases (NCDs) [6], mainly including cardio-cerebrovascular diseases, cancers, chronic respiratory diseases and diabetes [7]. Thus, the prevention of NCDs and the maintenance of HRQOL are major difficulties for the achivement of healthy aging. 

Previous studies have focused on the association between one specific NCD and HRQOL, such as hypertension [8], type 2 diabetes [9] and chronic kidney disease [10]. However, the results were not consistent, since most of these studies were cross-sectional studies. A lot of studies showed a negative effect of one specific NCD on HRQOL, while others did not find a correlation [8,9,10,11]. Additionally, most studies have ignored the influence of the number of NCDs, while older adults usually have more than one NCD, referred to as multimorbidity [12]. Some international researchers investigated the HRQOL of people suffering from multiple NCDs in their countries and found a negative relationship between HRQOL and multimorbidity [13,14]. However, few studies focused on older adults, especially Chinese older adults [15,16]. In fact, the relationship between the HRQOL and NCDs is very complex and numerous factors could make a difference. For instance, a great number of NCDs are due to unhealthy behaviors, especially poor diets [7]. A few researchers proposed the “modernization theory” to explain the rise of unhealthy diets [17]. The theory illustrated burgeoning middle classes with more disposable income tended to consume more meat and processed foods that had close associations with weight gain. Therefore, to some extent, it is reasonable to believe people with NCDs could have a good economic status. On the other hand, social demographic determinants exert a tremendous effect on HRQOL [18]. Among these factors, socioeconomic status has been proven to be an independent predictor of HRQOL [19]. People with higher socioeconomic status often have a better HRQOL [20]. Hence, it is reasonable to argue that NCDs might exert little or even no negative effect on HRQOL on account of the good economic status of people with NCDs. The relationship between HRQOL and NCDs should be further clarified through a large-scale survey.

In China, due to the rapid process of industrialization and urbanization, citizens’ lifestyle and disease spectrum have changed a lot [5]. Nearly 180 million Chinese older adult people suffer from NCDs, accounting for 75% of older people in China, and the prevalence of multimorbidity is about 35% [21]. Many studies on the HRQOL and NCDs among older people in China simply classified subjects according to whether they had NCDs or not, while few studies focused on the number of NCDs. Moreover, the samples of those studies were too small to be representative [15,22].

Therefore, we conducted the cross-sectional investigation to understand the HRQOL among people with different numbers of NCDs., and to explore the relationship between them. We argue that the HRQOL of Chinese older adults might become worse with the increase in the number of NCDs. The findings are expected to provide theoretical support for the timely advance of NCD management.

## 2. Materials and Methods 

### 2.1. Study Population

The cross-sectional study was based on a large-scale survey named China’s Health-Related Quality of Life Survey for Older Adults 2018 (CHRQLS-OA 2018) whose chief initiator and executor was the Global Health Institute of Wuhan University [23]. Interviewers also included teachers and students from other top universities in China (such as Nanchang University). All staff received professional training before the survey started. Conducted in either online or paper format between January and March of 2018, the survey recruited older adults aged 60 years and over across the country. Through convenience sampling, 5442 valid subjects were recruited. The survey mainly collected participants’ individual socio-demographic characteristics, social capital, behaviors and lifestyles, health-related quality of life, mental health and coping strategies, etc.

This study aims to explore the relationship between the number of NCDs and the HRQOL of Chinese older adults. In the present study, 276 subjects (5.07%) without information on NCDs and/or HRQOL were excluded. Finally, 5166 respondents (94.93%) were included in the analysis. 

### 2.2. Description of the Measures

#### 2.2.1. General Demographic Characteristics

In the present study, the following general demographic characteristics of the participants were included: age, sex, nationality, body mass index (BMI), years of education, household registration, marital status, average annual household income (CNY), new personal savings and self-rated health status (self-rated health status was based on individuals’ subjective feelings). The variable was divided into three groups, which were “Good”, “General”, and “Poor”. “General” was a status between “Good” and “Poor”, and was equivalent to “middle” or “moderate”. These general demographic characteristics were considered as confounding factors in the analysis of the cross-sectional association between the number of NCDs and HRQOL among Chinese older adults.

#### 2.2.2. Assessment of the Number of NCDs

In “China’s Health-related older adults Quality of Life Survey 2018”, the number of NCDs and the time of diagnosis were self-reported. Diseases investigated included hypertension, diabetes, coronary heart disease, tumor, chronic obstructive pulmonary disease (COPD), intervertebral disc disease, asthma, rheumatoid arthritis, gastroenteritis, cataract, cerebrovascular disease and others (openly filled in). Professional medical diagnoses can reduce certain deviation, and thus the questionnaire refered to the following questions: “where and by whom was the disease diagnosed? How long has the diagnosis been made?”

#### 2.2.3. Assessment of the HRQOL

The WHO’s HRQOL Research Group has further developed the World Health Organization Quality of Life-Old (WHOQOL-OLD) module for older adults based on the original QOL scale [24,25]. The WHOQOL-OLD module can be widely used in a variety of studies concerning crucial issues of the HRQOL in old age, such as population epidemiology, health monitoring, service development and so on. The WHOQOL-OLD module includes 24 items assigned to six aspects (each of the aspects has 4 items): the “Sensory Abilities” (SAB) aspect assesses sensory capacity and the influence of loss of sensory abilities on HRQOL; the “Autonomy” (AUT) aspect means independence in old age; the “Past, Present and Future Activities” (PPF) aspect assesses satisfaction with one’s achievements in life and expectations in the future; the “Social Participation” (SOP) aspect refers to participation in daily activities; the “Death and Dying” (DAD) aspect is related to concerns, worries and fears about death and dying; the “Intimacy” (INT) describes the ability to have personal and intimate relationships. The scores of each item range from 1 to 5, thus the scores of each aspect range from 4 to 20. The scores of all the six aspects can be combined into a total (“overall”) score for HRQOL in old age. Essentially, high scores represent high HRQOL, low scores represent lower HRQOL. Through the evaluation of the Chinese version of the WHOQOL-OLD module, the results showed good reliability and validity, thus WHOQOL-OLD is suitable for the HRQOL evaluation of Chinese older adults [26]. 

### 2.3. Statistical Analysis

The Statistical Package for the Social Sciences (SPSS) version 23.0 for Windows (SPSS Inc., Chicago, IL, USA) was applied to conduct all statistical analyses, with a statistical significance level of 0.05. The data analysis consisted of three aspects. First, frequencies and proportions were reported to describe the distribution of NCDs in the population with different demographic characteristics. Second, the difference in scores of all dimensions of the WHOQOL-OLD scale was examined among groups with different numbers of NCDs by ANOVA, and the Student–Newman–Keuls (SNK) test method was used for post-hoc comparisons. Third, multiple linear regression models were used to determine the relationship between the number of NCDs and the total scores and scores of each dimension of the WHOQOL-OLD scale. We established three models in the third step, including the initial model (model 1) and the adjusted models (model 2 and model 3). The final parsimonious model adjusted for confounders that were significantly associated with the dependent variables in model 2. The unstandardized coefficients (B) with a 95% confidence interval (95% CI) obtained from the models were reported.

### 2.4. Ethical Statements

This study was conducted in accordance with the Declaration of Helsinki, and the study protocol was reviewed and approved by the Institutional Review Board of School of Health Science and Faculty of Medical Sciences, Wuhan University (IRB number: 2019YF2050). Informed consent information was included in each questionnaire and introduced before the surveys. The questionnaires were completed by participants themselves or their caregivers according to older adults’s reading and response abilities. Surveys were only conducted when subjects were fully informed of the content and aim of this research project and agreed to participate. The survey was also conducted anonymously, and respondents’ information was kept confidential and only for the use of scientific research.

## 3. Results

### 3.1. Descriptions of Sample Characteristics

A total of 5166 older adults was included in the study. The general demographic characteristics are shown in Table 1. In general, there were 51.3% older adults with one or more NCD(s). More than half (50.9%) of men did not have NCDs, while only 46.6% of women did not have NCDs. Most people aged 60–64 years and those aged 85 years or older did not suffer from NCDs (54.0% and 55.3%, respectively). The Han population (50.8%) had a lower proportion of NCDs than minorities (58.5%). People with a normal BMI had the highest percentage of those without NCDs (50.8%). People with NCDs accounted for 58.2% of uneducated participants, which was higher than those receiving education. The proportion of non-agricultural people who did not suffer from NCDs (53.5%) was significantly higher than that of agricultural people (46.6%). People who were married or cohabiting had a lower percentage of having any NCDs (48.6%). Nearly three fifths (59.6%) of the population with an average annual household income of 45,001–60,000 did not suffer from NCDs. About two thirds (62.4%) of people who self-rated their health status as good did not suffer from NCDs. The proportion of people with new personal savings of 100,000 CNY or more who did not suffer from NCDs reached 63.1%. 

### 3.2. Description of the Scores of the WHOQOL-OLD Scale

As shown in Table 2, there were differences in total score and scores of the six dimensions of the WHOQOL-OLD scale among people with different numbers of NCDs, and the differences were statistically significant (*p* < 0.001). Those with the highest scores of SAB, AUT, PPF, SOP, DAD, INT and total score were people without NCDs, while those with the lowest scores were people with two NCDs or three NCDs and above. The multiple comparisons found that as the number of NCDs increased, the scores of each dimension and the overall score gradually decreased.

### 3.3. Relationship between the Number of NCDs and the Scores of the WHOQOL-OLD Scale

As shown in Table 3,multiple linear regression analysis was used to identify the relationship between the number of NCDs and the scores of the WHOQOL-OLD scale, and the model 2 and model 3 were adjusted by confounding factors. In the crude and adjusted models, the number of NCDs was related to the scores of the WHOQOL-OLD scale. According to the final model 3, with no NCDs as the reference group, having one NCD had a negative impact on the three dimensions of PPF, SOP, and DAD in the HRQOL of older adults (*p* < 0.05). Suffering from two NCDs had negative effects on SAB, AUT, PPF, SOP, and DAD scores of older adults (*p* < 0.05). Suffering from three or more NCDs exerted negative effects on all dimensions in the HRQOL of older adults (*p* < 0.05). In terms of the WHOQOL-OLD total score of people with different numbers of NCDs, as compared with the reference group (with zero number of NCDs), suffering from one NCD (B = –0.87, 95% CI = –1.67 to –0.08, *p* < 0.05), two NCDs (B = –2.89, 95% CI = –3.87 to –1.90, *p* < 0.001), and three or more NCDs (B = –4.20, 95% CI = –5.36 to –3.05, *p* < 0.001), all had negative impacts on the HRQOL of older adults, and as the number of NCDs increased, the HRQOL of older adults became worse, after adjusting for the confounding factors that were significantly associated with HRQOL.

## 4. Discussion

In the present study, we examined the association between the number of NCDs and HRQOL among Chinese older adults. Suffering from one NCD had a slight negative effect on the HRQOL of older adults, but as the number of NCDs increased, this negative effect also gradually increased.

The healthy aging proposed by the WHO is not only the extension of life, but also the improvement of the HRQOL of older adults. At the same time, NCDs serving as a negative factor of the HRQOL of older adults have received increasing attention [27]. In China, Huang’s [28] survey in Heilongjiang Province found that people with NCDs had a worse HRQOL than people without NCDs. A cross-sectional study of the HRQOL of older adults conducted by Zhu Yaxin [29] in Liaoning Province concluded that NCDs contributed to the poor HRQOL of older adults. However, these studies did not consider the impact of the number of NCDs on HRQOL. Previous studies from developed countries or regions have confirmed the relationship between NCDs and the HRQOL [11,30]. For instance, Wacker [31] evaluated the HRQOL of 2291 COPD patients, and the results showed that COPD combined with other NCDs significantly reduced HRQOL. These studies have almost proved the harmful effect of NCDs on the HRQOL. However, the relationship between the number of NCDs and the HRQOL is worth discussing.

In the results of this study, suffering from one NCD had a negative impact on the HRQOL of older people. However, when compared with two or more NCDs, this negative effect was relatively minor, which was consistent with Xin Yu’s survey results in southern China [16]. In fact, NCDs are generally considered as diseases with an unknown etiology, long duration, and a generally slow progression [32]. These characteristics may partly explain the smaller impact of having one NCDs on HRQOL. Moreover, related studies have found a positive correlation between economics and NCDs [33]. Therefore, it is reasonable to speculate that older adults with one NCD have a higher economic level, which may also alleviate the harmful effect of NCDs on HRQOL.

By further comparing the dimensions of HRQOL, the results of this study showed that the impact of having one NCD on three dimensions (SAB, AUT, and INT) of HRQOL was not significant. SAB, AUT, and INT evaluate the sensory ability, independent ability and intimacy of older people, respectively. However, NCDs generally refer to medical diseases with a long duration and no infection [7]. Therefore, the hypothesis that one NCD has a weaker effect on the sensory ability and independent ability of older adults is reliable. Moreover, NCDs are not contagious and have almost no external manifestations, which might explain why there was no difference in intimacy between people without NCDs and with one NCD. Unlike the above three dimensions, having one NCD had significant negative effects on the other three dimensions (PPF, SOP, and DAD) of HRQOL. PPF, SOP, and DAD evaluate the satisfaction with expectations, participation in social activity, and care, worry, and fear of death of older people, respectively. Older adults with NCDs not only experience a change in their physical health, but also bear a huge psychological burden [34]. This could partly explain why older people lower their expectations for the future. Moreover, because of the existence of NCDs, older people’s social participations are likely to be restricted [35]. Since older adults are at the end of the life cycle, they are already facing the fear of death [36], which will inevitably increase once they suffer from a disease. All in all, having one NCD had a slight impact on the HRQOL of older adults, which may be a potential hazard. It may cause older adults with one NCD to ignore the prevention and management of the disease, which may lead to multiple diseases and a worse HRQOL.

As our research results showed, suffering from two or more NCDs had a more serious negative impact on the HRQOL of older adults. The results are consistent with other studies. Previous research has shown that NCDs can impair the HRQOL of older adults, and the impairment was more severe when suffering from multiple NCDs, which may be related to the additive or synergistic damage caused by various NCDs to the body [37]. In China, Deng [38] found that adults with NCDs had a lower HRQOL than adults without NCDs, and their HRQOL had declined more significantly as the number of NCDs increased. Multiple studies in other countries have also demonstrated the inverse relationship between the number of NCDs and HRQOL [39,40]. Furthermore, we found that having two NCDs had a significant negative effect on the five dimensions (SAB, AUT, PPF, SOP, DAD) of the HRQOL of older adults. When older people had ≥three NCDs, all dimensions of the HRQOL of older adults were negatively affected. This further confirms the fact that, as the number of NCDs increases, the negative impact on the HRQOL of older adults also increases. Therefore, our study provided strong evidence for the fact that older adult people with multimorbidity might have a worse HRQOL.

It is worth noting that NCDs are considered to be complication-prone diseases, especially among older adults [41]. Although the harm of multimorbidity to older adults’ HRQOL has been proven in the previous studies, the incidence of multimorbidity has continued to rise in recent years, and the HRQOL of older adults remains pessimistic [42]. Therefore, the best option to improve the HRQOL of older adults is to attach importance to the primary prevention of NCDs, which has been proven to be effective [43,44]. For older adults with one NCD, more attention should be paid to the treatment and management of the disease to avoid greater harm to the body caused by comorbidities, and fundamentally improves the HRQOL.

Generally speaking, implementing primary prevention plays a key role in preventing the occurrence of NCDs and improving the HRQOL of older adults. However, we should also note that there was a close relationship between the number of NCDs and the HRQOL. As the number of NCDs increased, the HRQOL of older adults became worse. Therefore, we should pay attention to the prevention and management of NCDs of older adults to prevent the occurrence of multiple diseases.

Some limitations of this study should be taken into consideration. First, the existence of NCDs was self-reported by the respondents, which may lead to the omission of undetected diseases. Second, the types of NCDs listed in the questionnaire were incomplete, which could result in the omission of other NCDs. At the same time, due to the lack of data, the impact of NCD duration on HRQOL was not considered. Obviously, HRQOLs of patients with different types of NCDs are different, and people with the same disease may have a different HRQOL due to different treatment stages. However, considering the complex combination of disease types, in this study, we did not condider the impact of different diseases and their severity on HRQOL. Third, other potential confounding variables, such as smoking and drinking, were not adjusted in the model. Fourth, this cross-sectional survey does not allow us to obtain the dynamic impact of a single NCD or multiple NCDs on HRQOL. 

## 5. Conclusions

This study clearly showed the positive relationship between the number of NCDs and the HRQOL of older people. Compared with older adults without NCDs, suffering from one NCD had a slight negative impact on the HRQOL of older adults; however, having two or more NCDs exerted a greater negative effect on the HRQOL of older adults. As the number of NCDs increased, the HRQOL of older adults was worse. Therefore, we suggest primary prevention as the key to the improvement of HRQOL of older adults by avoiding or delaying the occurrence of one NCD. More importantly, considering the seeming harmlessness of one NCD and the negative effect of multimorbidity on HRQOL, more importance should be attached to joint interventions and programs that target the prevention and control of multimorbidity among older adults with one NCD.

## Figures and Tables

**Table 1 ijerph-17-05150-t001:** The number of noncommunicable diseases (NCDs) in different demographic groups.

Variables	The Number of Noncommunicable Diseases (NCDs)			
	0	1	2	≥3
	*n* = 2517(48.7%)	*n* = 1358 (26.3%)	*n* = 760 (14.7%)	*n* = 531 (10.3%)
Gender				
Male	1299 (50.9%)	681 (26.7%)	332 (13.0%)	238 (9.3%)
Female	1202 (46.6%)	668 (25.9%)	423 (16.4%)	289 (11.2%)
Age				
60–64	618 (54.0%)	269 (23.5%)	143 (12.5%)	114 (10.0%)
65–69	581 (48.5%)	321 (26.8%)	179 (14.9%)	118 (9.8%)
70–74	551 (46.2%)	352 (29.5%)	174 (14.6%)	116 (9.7%)
75–79	327 (44.2%)	203 (27.5%)	118 (16.0%)	91 (12.3%)
80–84	257 (45.2%)	141 (24.8%)	100 (17.6%)	71 (12.5%)
≥85	167 (55.3%)	70 (23.2%)	45 (14.9%)	20 (6.6%)
Nationality				
Han	2369 (49.2%)	1255 (26.1%)	690 (14.3%)	498 (10.3%)
Minority	102 (41.5%)	76 (30.9%)	44 (17.9%)	24 (9.8%)
BMI				
Underweight (<18.5)	253 (46.9%)	155 (28.7%)	81 (15.0%)	51 (9.4%)
Normal (18.5–23.9)	1601 (50.8%)	806 (25.6%)	438 (13.9%)	306 (9.7%)
Overweight (24–27.9)	479 (43.7%)	315 (28.7%)	177 (16.1%)	126 (11.5%)
Obese (≥28)	95 (41.3%)	57 (24.8%)	40 (17.4%)	38 (16.5%)
Years of education				
0	471 (41.8%)	313 (27.7%)	193 (17.1%)	151 (13.4%)
1–5	715 (49.4%)	377 (26.1%)	212 (14.7%)	142 (9.8%)
6–8	461 (49.7%)	241 (26.0%)	139 (15.0%)	87 (9.4%)
9–11	331 (48.0%)	182 (26.4%)	97 (14.1%)	80 (11.6%)
≥12	300 (49.6%)	180 (29.8%)	78 (12.9%)	47 (7.8%)
Household registration				
Agricultural	1626 (46.6%)	948 (27.1%)	534 (15.3%)	385 (11.0%)
Non-agricultural	852 (53.5%)	389 (24.4%)	214 (13.4%)	137 (8.6%)
Marital status				
Married/cohabiting	1687 (51.4%)	857 (26.1%)	433 (13.2%)	302 (9.2%)
others	815 (43.9%)	492 (26.5%)	323 (17.4%)	225 (12.1%)
Average annual household income (CNY)				
<15,000	816 (45.0%)	494 (27.3%)	274 (15.1%)	228 (12.6%)
15,000–30,000	552 (43.3%)	352 (27.6%)	215 (16.9%)	156 (12.2%)
30,000–45,000	523 (54.6%)	234 (24.4%)	135 (14.1%)	66 (6.9%)
45,001–60,000	334 (59.6%)	127 (22.7%)	59 (10.5%)	40 (7.1%)
>60,000	254 (53.6%)	135 (28.5%)	59 (12.4%)	26 (5.5%)
Self-rated health status				
Good	942 (62.4%)	347 (23.0%)	163 (10.8%)	57 (3.8%)
General	1268 (48.8%)	724 (27.9%)	384 (14.8%)	220 (8.5%)
Poor	294 (28.3%)	283 (27.2%)	211 (20.3%)	251 (24.2%)
New personal savings (CNY)				
<10,000	862 (43.5%)	507 (25.6%)	324 (16.4%)	288 (14.5%)
10,000–30,000	473 (45.1%)	301 (28.7%)	169 (16.1%)	106 (10.1%)
30,000–50,000	355 (53.5%)	169 (25.5%)	94 (14.2%)	46 (6.9%)
50,000–100,000	360 (49.7%)	213 (29.4%)	102 (14.1%)	49 (6.8%)
≥100,000	438 (63.1%)	157 (22.6%)	62 (8.9%)	37 (5.3%)

Note: Sample sizes of the demographic characteristic variables may not sum to *n* = 5166 due to missing values.

**Table 2 ijerph-17-05150-t002:** Scores of World Health Organization Quality of Life-Old (WHOQOL-OLD) among the population with different numbers of NCDs.

Domains	Score Range	The Number of Noncommunicable Diseases (NCDs)				*F*	*p*	Multiple Comparisons
		0 (g 1)	1 (g 2)	2 (g 3)	≥3 (g 4)			
Sensory Abilities (SAB)	4–20	13.60 ± 3.42	13.12 ± 3.56	12.10 ± 3.40	11.21 ± 3.62	89.376	*p* < 0.001	g 4 < g 3 < g 2 < g 1
Autonomy (AUT)	4–20	13.93 ± 3.34	13.74 ± 3.34	13.21 ± 3.21	12.91 ± 3.39	19.369	*p* < 0.001	g 4, g 3 < g 2, g 1
Past, Present and Future Activities (PPF)	4–20	13.50 ± 3.15	13.03 ± 2.94	12.57 ± 2.79	12.25 ± 2.99	36.733	*p* < 0.001	g 4, g 3 < g 2 < g 1
Social Participation (SOP)	4–20	13.37 ± 3.04	12.78 ± 2.92	12.64 ± 2.73	12.10 ± 2.94	35.342	*p* < 0.001	g 4, < g 3, g 2 < g 1
Death and Dying (DAD)	4–20	12.53 ± 3.20	12.09 ± 3.31	11.62 ± 3.57	11.51 ± 4.09	23.346	*p* < 0.001	g 4, g 3 < g 2 < g 1
Intimacy (INT)	4–20	12.95 ± 3.32	12.95 ± 3.32	12.67 ± 3.39	11.74 ± 3.87	20.042	*p* < 0.001	g 4 < g 3 < g 2, g 1
Total score	24–120	79.87 ± 13.25	77.71 ± 13.13	74.80 ± 12.49	71.73 ± 14.56	70.701	*p* < 0.001	g 4 < g 3 < g 2 < g 1

**Table 3 ijerph-17-05150-t003:** Relationship between the number of NCDs and WHOQOL-OLD scale scores.

Models	Sensory Abilities (SAB)	Autonomy (AUT)	Past, Present and Future (PPF)	Social Participation (SOP)	Death and Dying (DAD)	Intimacy (INT)	Total score
	Unstandardized Coefficients β (95% Confidence Interval for β)						
Model 1							
0	Reference	Reference	Reference	Reference	Reference	Reference	Reference
1	−	−0.19 (−041, 0.03)	−0.47 (−0.69, −0.27) ***	−0.59 (−0.78, −0.39) ***	−0.45 (−0.67, −0.22) **	−0.01 (−0.22, 0.22)	−2.17 (−3.04, −1.29) ***
2	−1.50 (−1.78, −1.22) ***	−0.72 (−0.99, −0.45) ***	−0.94 (−1.18, −0.69) ***	−0.72 (−0.96, −0.48) ***	−0.92 (−1.19, −0.64) ***	−0.28 (−0.56, −0.01) *	−5.08 (−6.15, −4.00) ***
≥3	−2.39 (−2.71, −2.06) ***	−1.02 (−1.33, −0.71) ***	−1.25 (−1.54, −0.97) ***	−1.26 (−1.54, −0.99) ***	−1.02 (−1.34, −0.70) ***	−1.21 (−1.52, −0.89) ***	−8.15 (−9.39, −6.91) ***
Model 2							
0	Reference	Reference	Reference	Reference	Reference	Reference	Reference
1	−0.01 (−0.21, 0.21)	0.12 (−0.19, 0.25)	−0.29 (−0.48, −0.10) **	−0.37 (−0.56, −0.18) ***	−0.26 (−0.49, −0.03) *	0.16 (−0.07, 0.38)	−0.68 (−1.48, 0.12)
2	−0.85 (−1.11, −0.58) ***	−0.43 (−0.66, −0.16) **	−0.58 (−0.82, −0.34) ***	−0.36 (−0.60, −0.13) **	−0.83 (−1.12, −0.54) ***	0.04 (−0.24, 0.32)	−2.91 (−3.91, −1.91) ***
≥3	−1.13 (−1.44, −0.82) ***	−0.56 (−0.86, −0.25) ***	−0.63 (−0.91, −0.35) ***	−0.57 (−0.84, −0.30) ***	−0.62 (−0.95, −0.28) ***	−0.71 (−1.03, −0.38) ***	−4.01 (−5.17, −2.84) ***
Model 3							
0	Reference	Reference	Reference	Reference	Reference	Reference	Reference
1	−0.02 (−0.23, 0.19)	0.06 (−0.15, 0.27)	−0.33 (−0.52, −0.14) ***	−0.42 (−0.61, −0.24) ***	−0.25 (−0.48, −0.02) *	0.18 (−0.04, 0.41)	−0.87 (−1.67, −0.08) *
2	−0.81 (−1.07, −0.55) ***	−0.33 (−0.59, −0.07) **	−0.58 (−0.81, −0.34) ***	−0.38 (−0.61, −0.15) ***	−0.80 (−1.08, −0.51) ***	0.04 (−0.23, 0.32)	−2.89 (−3.87, −1.90) ***
≥3	−1.19 (−1.50, −0.89) ***	−0.32 (−0.63, −0.02) **	−0.65 (−0.93, −0.37) ***	−0.59 (−0.86, −0.32) ***	−0.62 (−0.95, −0.29) ***	−0.72 (−1.04, −0.39) ***	−4.20 (−5.36, −3.05) ***

Note: * *p* < 0.05; ** *p* < 0.01; *** *p* < 0.001. Model 1 was the crude model. Model 2 adjusted for all the potential predictors (age, sex, nationality, BMI, years of education, household registration, marital status, average annual household income (CNY), new personal savings and self-rated health status). Model 3 was the final parsimonious model, adjusting for all the predictors that were significantly associated with HRQOL in Model 2.

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
