# Peer review of "Relationship between the Number of Noncommunicable Diseases and Health-Related Quality of Life in Chinese Older Adults: A Cross-Sectional Survey"

_ijerph, 2020, doi:10.3390/ijerph17145150_

Round 1
Reviewer 1 Report
No tengo sugerencias para los autores. Pensé que era un gran trabajo. La muestra es considerable para poder sacar estas conclusiones, se explica cómo se ha hecho y los resultados se entienden perfectamente.
Reviewer 2 Report
This study about The relationship between the number of noncommunicable diseases and health-related quality of life in Chinese elderly is interesting. However, there are some methodology concerns.
Major revisions:
- It has been presented the relationship between general demographic characteristics and number of noncommunicable diseases (Table 1). However, it is needed the information of the distribution and comparison of these variables with HRQoL scores. This way the multivariate models would be better understood as it is indicated that “Model 3 is the final parsimonious model, adjusted for all the predictors that were significantly associated with HRQOL in Model 2”. In this sense, it is needed to describe what adjustment variables were used in both models (Model 2 and Model 3) exactly. This way, the reader could see what variables have been selected when they are probably very correlated as e.g. “average anual household income” and “New personal savings”.
- How the self-rated health status was measured and which are its categories? “General” is a category of this variable? It appears in Table 1. On the other hand, self-rated health was included in the multivariate models? It is another way to measure the HRQoL and its inclusion could not to be adequate.
- What is the target population of this study? Please, describe the characteristics of the target population (size, location…) to evaluate the representativeness of the sample size.
- Some important information should be described in methodology: how and by whom the surveys were administered and about which diseases were asked. This way, the second limitation described in the Discussion section (“the types of NCDs listed in the questionnaire were incomplete, which could result in the omission of other NCDs.”) could be better understood.On the other hand, the duration of NCDs was a collected variable but it has not been described nor included in models.
- It is suggested to described in Results section the type and the distribution of the noncommunicable diseases.
- Discussion section.
- It is suggested that the first paragraph of this section to be a summary of the main results of the study.
- The second paragraph is a justification of doing the study again, similar than some paragraph of Introduction section.
- In the third paragraph, the explanation of the results with the economic status would not be adequate if economic variables as “average anual household income” or “new personal savings” were included in the multivariate models. If economic status is so important and taking into account the relationship explained in discussion, it would be interesting to test the modify effect of this variable in the association between NCDs and HRQoL.
- It is said: “Our research suggests that implementing NCD preventions, especially primary prevention, is the key to improving the HRQOL of the elderly.”. This is not a conclusion of the results of this study.
- Another limitation of this study to include in the discusion section would be that models had not been adjusted for other potential confounding variables as smoke, alcohol, diet, physical activity…
Minor revisions:
- It is suggested to write the complete name in the first apareance of words before the acronym, e.g. WHOQOL-OLD (World Health Organization Quality of Life-Old).
- It is suggested to name the section 2.3. Statistical Analysis.
- Please, indicate the test used to do the multiple comparison in ANOVA test.
- Please, describe the complete name of acronyms in Tables.
- Why is used the bold in the results of SAB score?
- Please, confirm the range of the categories of BMI variable. In which category would be a participant with 24 or 27?
- It is suggested to include the range of scores of each dimension and in the total score in Table 2 to better interpretation of results.
- Please, include the references after the phrase “However, few studies focused on the elderly, especially Chinese elderly.” in Introduction section. It is suggested to include what is the novelty of this study with respect to them.
Reviewer 3 Report
SCOPE: the manuscript is in line with the thematic scope of the IJERPH journal.
TITLE: should be more informative – it should identify if the study is a cross-sectional survey, systematic review, meta-analysis, replication study, ect. (see „Instructions for Authors”, please).
ABSTRACT: the length is correct, the abstract should not contain headers headings – see „Instructions for Authors”, please.
INTRODUCTION: too little attention seems to have been paid to the issue of NCDs – this is the main issue of this study, so it would be justified. Authors may consider replacing research questions with hypotheses that have been tested - this would introduce more clarity. Did the authors test hypotheses based on a specific theory of the relationship between NCDs and quality of life? Or is it only a descriptive study? It is worth formulating the research goal in the Introduction section.
MATERIALS AND METHODS: for older people who were illiterate, how could you get the written confirmation? Who distributed and collected questionnaires? How did you reach potential respondents (I know it happened during the Spring Festival)? Was it a team of trained interviewers? How was consent to participate in the study obtained from illiterate seniors? Properly selected questionnaire and statistical tests.
RESULTS: the results are consistent and logical.
DISSCUSSION: the information contained in lines 205-208 should appear in the abstract – this is part of the originality of this study.
CONCLUSIONS: it would be reasonable to formulate conclusions equal to the number of research questions (or hypotheses) in the Introduction section – a significant number of applications are postulative (lines 273-277).
REFERENCES: I recommend verifying the citation of the bibliography item (step by step) – see „Instructions for Authors”, please. Bibliography is current, most of bibliography items are after 2010.
GENERAL COMMENTS: manuscript is well written and clear. The word ‘elderly’ is the ageist language. It is recommend to replace ‘the elderly’ in this article to the terms like ‘older adults’, ‘elders’ or ‘older people’. This comment also applies to the ‘elderly’ in the title, keywords, and other places that use ‘the elderly’.
Author Response
Please see the attachmen.

Round 2
Reviewer 2 Report
Thanks for the opportunity to review this article. The manuscript has been improved. However, there are suggestions which have not been added or sufficiently justified to not do it.
- It continues being suggested to include the description of the categorization used in self-rated health status (line 91). Precisely, the section is called “Description of the measures”.
- Although the study aimed to explore the association between the number of NCDs and HRQOL, the readers could think that it is not the same to have one NCDs as cancer than cataract. It is therefore that it continues being suggested to include, briefly, the description of the type and the distribution of the noncommunicable diseases in Results section.
- Tables should be self-explained. It is therefore that it continues being suggested to include the range of scores of each dimension and in the total score in Table 2 to better interpretation of results.
- The conclusion about few studies focused on the related topic among Chinese elderly could be correct but it is necessary to include the few researches found as references after this affirmation to justify that in Introduction section (line 49).
Reviewer 3 Report
The quality of the article has improved, the comments of the reviewer have been taken into account. I recommend this article for publication in its present form.
